# Untangling the threads of cellulose mercerization

**Daisuke Sawada[1], Yoshiharu Nishiyama[2], Riddhi Shah[1], V. Trevor Forsyth[3,4,5], Estelle Mossou[3], Hugh Michael O'Neill [1], Masahisa Wada[6] & Paul Langan [1,7]**

Naturally occurring plant cellulose, our most abundant renewable resource, consists of fibers of long polymer chains that are tightly packed in parallel arrays in either of two crystal phases collectively referred to as cellulose I. During mercerization, a process that involves treatment with sodium hydroxide, cellulose goes through a conversion to another crystal form called cellulose II, within which every other chain has remarkably changed direction. We designed a neutron diffraction experiment with deuterium labelling in order to understand how this change of cellulose chain direction is possible. Here we show that during mercerization of bacterial cellulose, chains fold back on themselves in a zigzag pattern to form crystalline anti-parallel domains. This result provides a molecular level understanding of one of the most widely used industrial processes for improving cellulosic materials.

Cellulose is generated at the plasma membrane of plants by transmembrane synthase complexes that assemble glucose into chains of (1–4) poly (β-D glucan) that can be thousands of monomers long[1]. In all synthase complexes studied so far, glucose is added to the growing C4 end of a chain in the cytosol as the C1 end emerges from the other side of the membrane into a cell wall or pellicle. As several chains simultaneously emerge they self-assemble into hydrogen-bonded sheets of parallel chains that are stacked on top of each other through hydrophobic forces and van der Waals interactions to form a crystalline elementary fibril. Elementary fibrils are therefore polar with distinct C1 and C4 ends[2], and in plants typically further assemble into hierarchical structures with other elementary cellulose fibrils in which it is possible that fibrils of opposite polarities can be side-by-side. Elementary fibrils have been found to adopt two crystal phases, $I_\alpha$ and $I_\beta$, which are collectively referred to as cellulose I[3]. $I_\alpha$ and $I_\beta$ both have unit cells containing parallel chains and differ mainly in the relative stagger of the hydrogen-bonded sheets along the chain axis direction[4,5].

In the mid-1800s, the chemist John Mercer discovered that treating fabrics woven from cellulosic cotton fibers with sodium hydroxide (NaOH) solutions improved their mechanical properties,

made them more receptive to dye, and gave them an attractive shiny appearance. Mercerization is still widely used today to enhance the properties of textiles and twines made from various naturally occurring cellulosic plant fibers. Regardless of the exact proportions of cellulose I allomorphs present, mercerization converts elementary fibrils into another crystal form called cellulose II[6–8]. Cellulose II has a unit cell that contains two antiparallel chains. Fibers of cellulose II, therefore, have no polarity and their ends are chemically identical with equal amounts of C1 and C4 chain ends[2]. Two main theories have been put forward to explain how this could happen.

In electron microscopy studies of algal cellulose that had been mercerized, Chanzy and Roche saw morphologies that were similar to those previously seen during the crystallization of synthetic polymers[9]. These so-called "shish-kebabs" are formed when long polymer chains fold back on themselves in a zigzag pattern to form crystalline domains of antiparallel chains. By analogy, Chanzy and Roche proposed that as elementary cellulose fibers are swollen by the penetration of NaOH solutions it is possible that individual chains become free enough to fold back on themselves to form antiparallel crystalline domains of cellulose II.

[1]Biology and Soft Matter Division, Oak Ridge National Laboratory, Oak Ridge, TN 37831, USA. [2]Centre de Recherches sur les Macromolécules Végétales – CNRS, Affiliated with the Joseph Fourier University of Grenoble, BP 53, 38041 Grenoble Cedex 9 France. [3]Life Sciences Group, Institut Laue-Langevin & Partnership for Structural Biology, 38042 Grenoble Cedex 9 France. [4]Faculty of Medicine, Lund University, 22184 Lund, Sweden. [5]LINXS Institute of Advanced Neutron and X-ray Science, Scheelevagen 19, 22370 Lund, Sweden. [6]Division of Forest and Biomaterials Science, Graduate School of Agriculture, Kyoto University, Kitashirakawa Oiwake-cho, Sakyo-ku, 606-8502 Kyoto, Japan. [7]Institut Laue-Langevin, 38042 Grenoble Cedex 9 France. ✉e-mail: langan@ill.fr

An alternative theory suggested by Kolpak and Blackwell is that during swelling the NaOH solution breaks the hydrogen bonds within elementary fibrils leaving hydrophobic stacks of parallel chains that are relatively free to move[10]. Although all stacks of chains from a single elementary fibril are parallel, those from neighboring fibrils may have the same or opposite polarity. As NaOH is removed, stacks of chains from one elementary fibril may aggregate with stacks with an opposite polarity that has migrated from another, thus forming crystalline domains of cellulose II. Differentiating between these two possibilities has been an outstanding challenge in cellulose research.

An important difference between these two theories is the origin of the chains that generate crystalline domains of cellulose II. In the theory of Chanzy and Roche, the two antiparallel chains that form the cellulose II unit cell originate from within the same elementary fibril, whereas in that of Kolpak and Blackwell they originate from neighboring elementary fibrils (see Fig. 1). One of the major techniques that has been applied over the years to examine the crystal structure of cellulose is X-ray fiber diffraction. However, with this technique, we have not found it possible to determine whether the chains originate from the same or from different elementary fibrils.

In this work, we addressed the question of whether the model of Chanzy and Roche or Kolpak and Blackwell is correct, by using neutron fiber diffraction in combination with deuterium labeling (replacing hydrogen, H, by its isotope deuterium, D). H and D have very different neutron scattering properties. We have developed a method for fully deuterating the entire fibril (referred to as a nanocrystal) produced by the bacterium *Glucanacetobacter xylinus*. We further have developed a technology to prepare well-aligned oriented samples based on bacterial nanocrystals so that the fiber symmetry of the samples is ensured. By examining data collected from a sample consisting of a 50:50 mixture of hydrogenous and deuterated nanocrystals we are able to conclude that during mercerization of bacterial cellulose, structure conversion occurs within individual nanocrystals according to the model of Chanzy and Roche. This result provides a molecular-level understanding of a widely used industrial process and helps resolve a long-standing problem.

## Results

*G. xylinus* is a bacterium that produces a cellulosic pellicle when grown on a simple aqueous medium that contains glycerol and salts. We developed a method of extracting cellulose I nanocrystals from these pellicles and then assembling them into well-aligned samples for fiber diffraction studies, as described in the methods section. We also developed a protocol for growing the bacteria on a medium in which all H had been replaced by D so that the resulting pellicle contained fully deuterated cellulose, as rigorously confirmed by Fourier transform infrared (FTIR) and mass spectrometry[11,12]. Aligned samples were prepared from hydrogenous nanocrystals, deuterated nanocrystals, and an equal (50:50) mixture of hydrogenous and deuterated nanocrystals, referred to as CH-cellulose I, CD-cellulose I, and CDH-cellulose I, respectively, and then after mercerization as CH-cellulose II, CD-cellulose II, and CDH-cellulose II.

X-ray fiber diffraction patterns from CH-cellulose II, CD-cellulose II, and CDH-cellulose II contain diffracted peaks with similar relative intensities, because H and D have similar (and relatively small) X-ray form factors. Neutrons interact with the nuclei of atoms through the strong interaction. H atoms have a relatively strong and negative scattering length (−3.74 fm) which means that they appear as negative troughs in nuclear density maps. On the other hand, the coherent neutron scattering length of D is positive and even larger in magnitude (+6.67 fm). Neutron fiber diffraction data, collected using previously described techniques[13] from CH-cellulose II, CD-cellulose II, and CDH-cellulose II, contain diffracted intensities with very different relative intensities because H and D have very different neutron scattering lengths (see Fig. 2). As discussed below, analysis of diffraction from the CDH-cellulose II sample enabled us to differentiate between the Chanzy and Roche and the Kolpak and Blackwell model, whereas diffraction from the other two samples, CH-cellulose II and CD-cellulose II, provided supporting evidence.

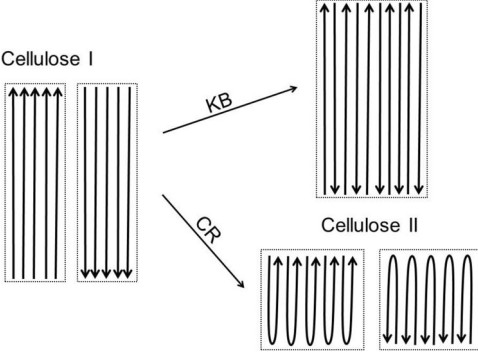

**Fig. 1 | Schematic representation of two mechanisms proposed for converting cellulose from a parallel to an antiparallel chain arrangement during mercerization.** Individual fibrils of cellulose I, or crystalline domains of cellulose II, are outlined by broken black lines. The chain direction is represented by an arrow. Two side-by-side elementary fibrils of cellulose I of opposite polarity are represented on the left-hand side. Within each fibril of cellulose I all chains are parallel. During swelling with NaOH, Kolpak and Blackwell proposed (KB arrow) that chains that are relatively free to move from one elementary fibril may aggregate with chains of opposite polarity that have migrated from another, resulting in the merging of fibrils into crystalline domains of cellulose II, as represented on the top right. On the other hand, Chanzy and Roche proposed (CR arrow) that individual chains become free enough to fold back on themselves, resulting in the formation of crystalline domains of cellulose II within individual fibrils of cellulose I, as represented on the bottom right. Differentiating between these two possibilities has been an outstanding challenge in cellulose research.

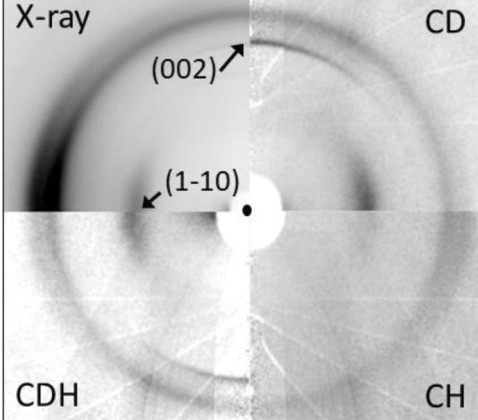

**Fig. 2 | Fiber diffraction patterns collected from cellulose I nanocrystals after mercerization to cellulose II.** The figure is a composite of quadrants from 4 patterns. Top left quadrant: using X-rays. Top right quadrant (CD): using neutrons with H in all nanocrystals replaced by D. Bottom right quadrant (CH): using neutrons with no D present. Bottom left quadrant (CDH): using neutrons with H in half of the nanocrystals replaced by D (CDH). The black circle in the center of the image represents the position of the direct beam as it passes straight through the sample. The vertical, meridional, direction corresponds to the direction of the fiber axis and the cellulose chains. The equator of the diffraction pattern passes horizontally through the beam center and corresponds to the packing direction of neighboring cellulose chains. The position of the (002) meridional and (1–10) equatorial diffraction intensities is indicated by arrows. The relative differences between the diffracted neutron intensities in (CD), (CH), and (CDH) are due to different distributions of H and D within the samples. The pattern of white lines in these neutron diffraction patterns is due to the way the data are folded together from several different detector positions, as described previously[13].

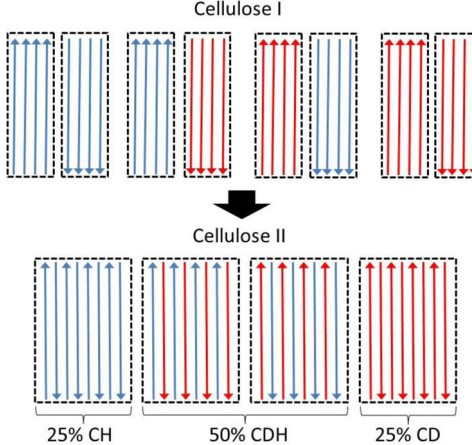

**Fig. 3 | Schematic representation of how diffracted neutron intensities from CDH-cellulose II were calculated based on the model of Kolpak and Blackwell.** We represent the case when conversion to cellulose II occurs only when chains move between neighboring cellulose I nanocrystals with opposite polarity. Individual nanocrystals in cellulose I, or crystalline domains in cellulose II, are outlined by broken black lines. Within crystalline domains, hydrogenous and deuterated cellulose chains are represented in blue and red, respectively. The chain direction is represented by an arrow. The top half of the diagram represents the four different possible interactions between hydrogenous and deuterated nanocrystals with opposite polarity. There are equal (50:50) amounts of hydrogenous and deuterated nanocrystals randomly distributed within cellulose I samples and with random up or down orientations with respect to the fiber axis. After mercerization, 50% of crystalline domains will contain equal amounts of hydrogenous and deuterated chains, 25% will contain just hydrogenous chains, and 25% will contain just deuterated chains. KB-calculated-neighbor is obtained from the sum of these contributions. The size of the unit cells for 25% CH and 25% CD and the averaged cell for 50% CDH is the same. Therefore, the position of intensities in the diffraction pattern from a sample containing mixtures of these different unit cells would coincide. We do not expect additional reflections.

Table 1 | Relative observed and calculated intensities of peaks (0 0 2) and (1 –1 0) from the neutron fiber diffraction patterns of cellulose II, normalized so that (0 0 2) is the unity

| | 0 0 2 intensity | 1 –1 0 intensity |
|---|---|---|
| Experimental: | | |
| CDH-cellulose II | 1.00 (5) | 4.28 (5) |
| CH-cellulose II added to CD-cellulose II | 1.00 (5) | 4.38 (7) |
| Calculation: | | |
| KB-calculation-statistical | 1 | 11.49 |
| KB-calculation-neighbor | 1 | 5.90 |
| CR-calculation | 1 | 4.11 |

The value in brackets is the error in the last digit. After this normalization, the intensity of (1 –1 0) is equivalent to the ratio of the intensity of (1 –1 0) to (0 0 2).

The size and space group of the unit cell, and therefore the position of diffraction intensities, did not vary significantly between the three different cellulose II samples. There is therefore no change in the number of chains or their relative arrangement within the unit cell resulting from processing the samples with different proportions of H or D. We measured the intensities of diffraction peaks for Bragg reflections (0 0 2) on the meridian (at -0.194 Å⁻¹ in reciprocal space) and (1 –1 0) on the equator (at -0.139 Å⁻¹ in reciprocal space) of the diffraction pattern from CDH-cellulose II, as described in the methods sections. The ratio of the intensities of these two low-resolution reflections varies significantly between the different cellulose II samples.

We compared the ratio of the measured intensities from CDH-cellulose II, with the ratio of intensities calculated based on the theory of Chanzy and Roche (the CR calculation), and the theory of Kolpak and Blackwell (the KB calculation). An important assumption for these calculations is that there are equal amounts of hydrogenous and deuterated nanocrystals randomly distributed within CDH-cellulose II samples and with random parallel or antiparallel orientations with respect to the fiber axis.

For the KB calculation, we considered two bounding cases which we believe represent two extremities of a broad range of possibilities. In each bounding case, during mercerization, the cellulose chains retain their orientation with respect to the fiber axis, but can move orthogonal to it. The two bounding cases differ in the freedom of chains to move in this lateral direction.

In the first bounding case, the lateral movement of chains is unhindered and during mercerization, a chain can move throughout the sample, freely interacting with any other chain of opposite polarity from any other nanocrystal. In cellulose II crystal domains that form after this complete mixing, any chain position will be randomly occupied by either a deuterated or hydrogenous chain in a statistically disordered manner. Analogous types of statistical disorder have been described in X-ray studies of fibers of DNA and cellulose and result in Bragg diffraction from an average or "statistical" crystal[14,15]. In our case, the calculated intensity from such a statistical crystal, which we refer to as KB-calculation-statistical, can be calculated from an averaged unit cell within which, at the position of each of the two antiparallel chains, there is a superposition of a hydrogenous chain and a deuterated chain each with half occupancy. We would also expect to see disordered scattering intensity originating from the difference between the scattering density of the statistical crystal and individual chains at any position within the real crystal, but we do not.

In the second bounding case, we consider the lateral movement of chains to be limited to between directly neighboring nanocrystals of opposite polarity, which we refer to as KB-calculation-neighbor. Here, four different combinations of hydrogenous and deuterated nanocrystals would be possible during mercerization, as shown in Fig. 3. The KB-calculation-neighbor is obtained by adding 50% of the calculated intensity from a unit cell with equal amounts of hydrogenous and deuterated chains, to 25% of the calculated intensity from a unit cell within which there are just hydrogenous chains, and to 25% of the calculated intensity from a unit cell within which there are just deuterated chains.

In the case of the CR calculation, the situation is much simpler, as conversion occurs within independent nanocrystals. The CR calculation is obtained by adding 50% of the calculated intensity from a unit cell within which there are hydrogenous chains, to 50% of the calculated intensity from a unit cell within which there are deuterated chains.

We find that the CR-calculation is in significantly better agreement with the data than either the KB-calculation-statistical or the KB-calculation-neighbor (see Table 1). Further, if Chanzy and Roche are correct then the observed neutron fiber diffraction pattern produced by CDH-cellulose II should equal the average of the two patterns produced by CH-cellulose II and CD-cellulose II, i.e., CDH-cellulose II and (CH-cellulose II + CD-cellulose II)/2 should both contain equal numbers of unit cells within which all of the cellulose chains are hydrogenous or deuterated. That is indeed the case, with the exception of the very low-angle region on the equator (see Fig. 4).

## Discussion

The results of the work indicate that during mercerization of bacterial cellulose, structure conversion occurs within single nanocrystals: i.e., chains fold back on themselves to form crystalline antiparallel

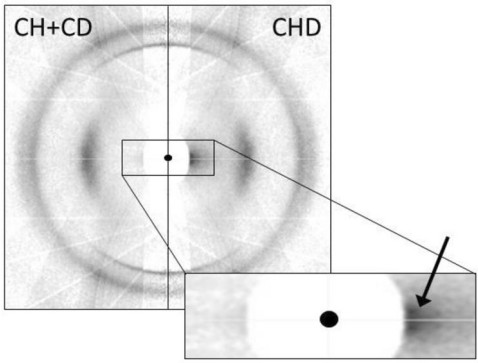

**Fig. 4 | Fiber diffraction patterns collected from cellulose II using neutrons.** The left half of the image (CH + CD) is the average of data collected from CH-cellulose II and CD-cellulose II. The right half of the image (CDH) is data collected from CDH-cellulose II. The black circle in the center of the image represents the position of the direct beam of neutrons after passing through the sample. The vertical, meridional, direction corresponds to the direction of the fiber axis and the cellulose chains. The equator of the diffraction pattern passes horizontally through the beam center and corresponds to the packing direction of neighboring cellulose chains. The two sides of the composite image are identical, with the exception of some low-angle intensity on the equator of CDH-cellulose II, indicated by an arrow in the enlarged section.

domains. Interaction of Na cations with cellulose must disrupt intra-chain hydrogen bonding in anhydrous cellulose I, allowing sufficient water to penetrate and swell the fibrils, and permitting cellulose chains the space and flexibility required to fold back on themselves. This result provides a molecular-level understanding of a widely used industrial process and helps resolve a long-standing problem.

There are caveats to this result. We cannot tell from this experiment if a chain folds back on itself just once so that it interacts with itself and neighboring chains in an antiparallel fashion, as schematically represented in Fig. 1, or if a chain folds back on itself several times in an accordion-like arrangement. Further, this experiment provides no information on the conformation of the cellulose chains in regions where they fold back on themselves and the extent to which these regions may be relatively large and disordered. However, we note that there is strong evidence for regular, and sharp, chain folding in the anomalous band-like cellulose II produced by the mutant bacterium (ATCC23769)[16].

In the Chanzy and Roche model, crystalline domains of cellulose II form within individual fibrils of cellulose I, rather than through the merging of fibrils of cellulose I as in the Kolpak and Blackwell model. These nanoscale domains of either hydrogenous or deuterated material could provide sufficient contrast to be observed using small-angle neutron scattering. We note that their aspect ratio will be different from the crystal domains in cellulose I, having decreased along the length of the cellulose chain direction and increased in the plane normal to this direction. For this reason, the intensity observed on the equator of diffraction from CDH-cellulose II at a low angle (Fig. 4) is intriguing. However, given the potential presence of large disordered domains of cellulose, there could be other explanations for this feature, and we have not conducted quantitative small-angle scattering studies.

Finally, it is important to note that native cellulosic materials are diverse and can contain cellulose fibers that differ significantly in size; from a few nanometers in thickness in plant cell walls to tens of nanometers in thickness in bacterial pellicules[17]. Furthermore, they can differ in the hierarchical arrangement of cellulose fibers and the presence of other molecular components, such as hemicellulose and lignin. It is possible, that mercerization of different native cellulosic materials may proceed through different molecular mechanisms.

## Methods

Nanocrystalline cellulose was produced using the following protocol. The bacterial strain *A. xylinus subsp. Sucrofermentans* (ATCC 700178) was purchased from the American Type Culture Collection (Manassas, Virginia, USA) and grown in defined growth media in order to produce either hydrogenous or deuterated cellulose pellicules, as rigorously confirmed by FTIR and mass spectrometry and as previously described[11,12]. Never-dried pellicules were suspended in water and centrifuged. HCl was added to a concentration of 4 mol/L and the suspension stirred for 4 h. The resulting suspension of cellulose nanocrystals was washed several times with distilled water, and sonicated.

We prepared three different suspensions in this way, each containing 45 mg of either hydrogenous nanocrystals, deuterated nanocrystals, or equal amounts of hydrogenous and deuterated nanocrystals, by dry weight. To each suspension, 100 mg of fibrinogen, and then a concentrated solution of thrombin, purchased from Sigma Aldrich, was added to produce a gel at room temperature. The gels were cut into strips of ~2 cm² in area and ~5 mms thick and stretched in one direction to about three times their original length, to align nanocrystals around the direction of stretching, i.e., the fiber axis. When left to dry under tension, these stretched 3 cm long strips shrunk in thickness to around 1 mm.

The aligned samples were then mercerized in solutions of 6 M NaOD/D$_2$O to completely convert from cellulose I to cellulose II, as verified by X-ray fiber diffraction. We used deuterated solutions to replace H associated with labile cellulose hydroxyl groups with D, in order to reduce unwanted neutron scattering background from the large incoherent neutron scattering produced by H. The intensities of OH and OD bands in FTIR spectra were used to verify substitution in each sample. We found that two cycles of mercerization treatment were sufficient to also remove all of the fibrinogen, as verified by the disappearance of a clear amide II band at 1550 cm$^{-1}$ in the FTIR spectra. The intensity of OH and OD groups in FTIR spectra indicated that the level of substitution was the same in each sample.

Samples for neutron fiber diffraction experiments were prepared as described above either using hydrogenous cellulose, deuterated cellulose, or equal amounts of hydrogenous and deuterated cellulose, referred to here as CH-cellulose II, CD-cellulose II, and CDH-cellulose II respectively. Available neutron beams are relatively weak compared to X-ray beams and therefore larger sample volumes are required to obtain sufficient scattering signal. In order to achieve sufficient sample volumes for our neutron studies, we assembled 2 mm thick parallel arrays of over 50 X-ray samples for each neutron experiment.

Neutron diffraction data were collected using previously described techniques[13]. We used the previously reported unit cell parameters of cellulose II (2 symmetry-independent antiparallel chains in space group $P2_1$ with $a = 8.01$ Å, $b = 9.04$ Å, $c = 10.35$ Å, $\gamma = 117.1°$) to index reflections in the pattern from CDH-cellulose (II)[5]. The two antiparallel cellulose chains are aligned along $c$ unit cell axis, and the plane orthogonal to the $c$ axis made by the $a$ and $b$ axes, contains the directions in which cellulose chains laterally pack together.

The relative intensities of equatorial (1 −1 0) and meridional (0 0 2) observed diffraction peaks in the CDH-cellulose II diffraction pattern were fitted by a Gaussian peak after background subtraction. The meridional reflection (0 0 2) corresponds to a d-spacing of 5.16 Å (distance between diffracting planes in real-space) and originates from neutron scattering density features in the unit cell projected onto the $c$ axis. The equatorial reflection (1 −1 0) corresponds to a d-spacing of 7.21 Å and originates from neutron density features projected onto the $a$ − $b$ direction.

**Table 2 | Calculated intensities of reflections (002) and (1 −1 0) for a unit cell containing only H chains, $I_H$(hkl), only D chains, $I_D$(hkl), and containing equal amounts of H and D chains (each with half occupancy at the two chain positions in the unit cell), $I_{DH}$(hkl), using the previously reported atomic coordinates of cellulose II[5] and the program SHELX[17]**

|          | 0 0 2 intensity | 1 −1 0 intensity |
|----------|-----------------|------------------|
| $I_H$    | 1228            | 9888             |
| $I_{DH}$ | 1862            | 21,392           |
| $I_D$    | 10,447          | 38,111           |

The neutron diffraction patterns contained background dominated by contributions from the incoherent scattering from H and D. Diffraction patterns from different samples were normalized with a scale factor obtained by fitting the measured incoherent scattering intensity, $I_o$, to the calculated value, $I_c$. $I_c$ was calculated from Eq. (1)

$$I_c = {I_o}/{e^{-\mu x}} \tag{1}$$

Where $\mu$ is the linear absorption coefficient calculated from Eq. (2)

$$\mu = \rho \sum_j (A)_j w_j \tag{2}$$

Where $\rho$ is the overall mass density of the sample, $A$ is mass absorption coefficient ($cm^2/g$) defined by wavelength, $w$ is the weight fraction of the $j$th atomic component in the sample, and $x$ is sample thickness (0.2 cm).

The intensities of each reflection, $I$(hkl), were calculated for a unit cell containing only H chains, $I_H$(hkl), only D chains, $I_D$(hkl), and containing equal amounts of H and D chains (each with half occupancy at the two chain positions in the unit cell), $I_{DH}$(hkl), using the previously reported atomic coordinates of cellulose II[5] and the program SHELX[18] (Table 2). The neutron coherent scattering lengths used in these calculations were 6.65 fm, 5.80 fm, −3.74 fm, and 6.67 fm for C, O, H, and D, respectively. As described in the Results section, the calculated values of $I$(002) and $I$(1−10) for different theories are obtained from Eqs. (3–5)

$$\text{KB-calculation-statistical} = I_{DH}(hkl) \tag{3}$$

$$\text{KB-calculation-neighbour} = I_{DH}(hkl)/2 + (I_H(hkl) + I_D(hkl))/4 \tag{4}$$

$$\text{CR-calculation} = (I_H(hkl) + I_D(hkl))/2 \tag{5}$$

In order to minimize the effect of any scaling errors, we used the ratio, $R$, of $I$(1−10)/$I$(002) for comparing observed and calculated data in Table 1, where $R_H = I_H(1−10)/I_H(002)$, and likewise for $R_D$ and $R_{DH}$.

For each calculated reflection intensity in Table 2, as the proportion of D compared to H increases, $I_H(hkl) < I_{DH}(hkl) < I_D(hkl)$. However, $R_D < R_H < R_{DH}$, which indicates that changing the proportion of D compared to H has a different effect on the values of $I$(1−10) and I(002), due to the spatial distribution (the structure) of H and D within the unit cell.

## Data availability
The data that support the findings of this study are available from the corresponding author upon request.

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

## Acknowledgements

This research is funded by the Genomic Science Program, Office of Biological and Environmental Research, U.S. Department of Energy, under FWP ERKP752. The Center for Structural Molecular Biology (CSMB) is supported by the Office of Biological and Environmental Research, using facilities supported by the U.S. Department of Energy, managed by UT-Battelle, LLC under contract no. DE-AC05-00OR22725. This research was supported by the Scientific User Facilities Division, Office of Basic Energy Sciences, US Department of Energy. The authors thank the Institut Laue Lagnevin for the provision of beamtime.

## Author contributions

D.S., R.S., H.M.O.N., and M.W. contributed to sample preparation. D.S., Y.N., V.T.F., E.M., and P.L. contributed to data collection. D.S. and P.L. contributed to the manuscript preparation. D.S. contributed to data analysis. D.S. and P.L. contributed to the design of the experiment.

## Competing interests
The authors declare no competing interests.
