## [Peer Review File · Nature Communications]

Untangling the Threads of Cellulose MercerizationReviewers' Comments:

Reviewer #1:

Remarks to the Author:

The manuscript entitled "Untangling the Threads of Cellulose Mercerization" by D. Sawada et al. addresses a very fundamental question in cellulose chemistry. Since the discovery of the crystallographic differences of native cellulose I and mercerized cellulose II, which would also be the crystal structure present in regenerated cellulose, it remained disputed how the transition from nanocrystals with parallel to those with antiparallel cellulose molecules would occur. A certain mobility of the long chain molecules is required and, in the case of mercerization, enabled by swelling with sodium hydroxide. Two options have been proposed for the rearrangement of the cellulose molecules. Chanzy and Roche (CR) suggest a folding mechanism of the chains within each crystal, whereas Kolpak and Blackwell (KP) assume migration of molecules from one crystal to another.

The challenge to decide whether the CR or the KB model is correct involves "marking" the molecules of one nanocrystal with respect to another one. The marker used in this work is an exchange of hydrogen by deuterium within the entire nanocrystal. The technique that deuterium exchange marking works for is neutron (fiber) diffraction. Deuteration can be achieved for bacterial cellulose: the bacterium *Glucanacetobacter xylinus* manages to live and grow in a fully deuterated environment. The authors further have developed a technology to prepare well-aligned oriented samples based on nanocrystals so that the fiber symmetry of the samples is ensured. The crucial sample is the one where a 50:50 mix of hydrogenated and deuterated nanocrystals is present; fully protonated and fully deuterated samples are measured for comparison.

For mercerization, a deuterated solution of NaOH (thus, NaOD + D₂O) was used. The authors carefully checked that the level of OH/OD exchange is similar in all samples; I think that could be stressed even more as the contrast technique is very sensitive to that.

For the logic of the work presented here, Fig. 1 is central. It is used in two ways. First, considering only the figure caption, the color scheme is more or less redundant as blue and red stand for two opposite orientations of the (chiral) cellulose molecules, depicted also by arrows. Second, in the main text, it is stated that blue and red may also signify protonated and deuterated chains in nanocrystals. Here, Fig. 1 does not show the full picture though. The starting point of the mixed sample before mercerization would be characterized by equal amounts of protonated and deuterated (thus, blue and red) nanocrystals with arrows up and down. If, in a simplified approach, the combination of chains of opposite polarity was only happening between neighboring nanocrystals of opposite polarity four different combinations of the blue and red arrows would be possible. Please refer to my attached sketch of the situation. Eventually, only 50 % of the cellulose II nanocrystals would still be mixed red and blue then, 25 % each would be just red or blue, i. e., fully protonated or fully deuterated, respectively. [Only if the chains were mobile beyond nearest neighbors, more complex combinations would also occur; see the rightmost figure in my sketch.] My feeling would be that the difference in Bragg reflection intensity ratios would be smaller than in Table 1. I cannot fully exclude that a decision between the CR and the KB model may no longer be possible on the basis of the measured data. I suggest that the authors make the according calculations to decide.

It is still valid though that the CR model would lead to the CDH pattern being an average of CH and CD. Still, I am not sure whether the corrected KB pattern would still be as distinctively different as Table 1 suggests. The nice additional argument via the pronounced small-angle scattering of the mixed sample is also valid, however, the data are not sufficiently quantitative here.

A last remark concerns the figures. It might be easier for the reader if, instead of ABCD, either the abbreviations of the model (CR and KB for Fig. 1) were used, in Fig. 2 X-ray, CH, CD, CDH may guide the reader, or in Fig. 3 CD+CH and CDH, respectively.

In conclusion, I hope that the innovative approach presented here in order to reveal the molecular origin of mercerization still works after the - in my opinion - necessary recalculation for the KB model.

2

3

4

Reviewer #2:

Remarks to the Author:

The manuscript describes a new approach to resolve a long-standing issue of the physical process behind mercerization of cellulose I to cellulose II. The authors were able to generate (sufficient) batches of hydrogenated and deuterated cellulose for a neutron diffraction experiment. By combining H-containing cellulose fibrils with D-containing cellulose fibrils, the authors perform a neutron diffraction experiment to discern between two models (CR vs KB) that describe the transition between the crystallographic state I and II. The approach is novel and the topic is relevant to bio-polymer physics in general, with cellulose as the most bio-abundant polymer.

The methodology is novel and clear but in interpreting the data, the authors can provide the reader with some assistance in reading the figures. The Q values of the reflections are assumed common knowledge as are the inter-domain and intra-domain ordering of the polymers.

The authors point out:

267: A detail of the lower angle region shows an equatorial intensity feature (indicated by arrow D) for CDH-cellulose II that does not appear for CH-cellulose II or CD-cellulose II.

The authors relate this 'strong feature' to small-angle scattering on nanoscopic crystalline domains: the required contrast (H-domain vs D-domain) would then also appear in their cellulose I samples, prior to mercerization, but such (supporting) data is not shown. Indeed, the observation itself calls for a SANS experiment.

Does the Q-value of the reflection corresponding to the domain size in an integer fashion? Figure 7c in [13] shows a similar darkening close to the beam stop, suggesting a similar ordering in 100%-D cellulose II, which would be in contradiction with the explanation given in the current manuscript. In the (current) manuscript, figure 2C shows a very faint darkening at the same Q value. Although one could discard such darkening as too faint, the variations in table 1 between integrated intensities are of the order of their inaccuracies (assumed to be stated in parentheses behind the intensity value).

some minor remarks/observations:

line 85: mixture of hydrogenated and deuteration nanocrystals,  deuterated?

line 243: pattern

The main conclusion of the manuscript is that experiment supports the C&R model rather than the K&B model, so polymer folds onto itself rather than intermixing with other molecules through diffusion. Table 1 shows that CDH-II is indistinguishable from the addition of CH-II and CD-II. The K&B process would result in a doubling of the cell parameter for the CHD-II system. Does the calculated intensity of table I correspond to the new cell reflection?

Summarizing, the manuscript provides new insights in an important bio-chemical process, but can easily be improved in discussing the interpretation in a more quantitative manner.

Point-by-point response

Reviewer #1 (Remarks to the Author):

The manuscript entitled "Untangling the Threads of Cellulose Mercerization" by D. Sawada et al. addresses a very fundamental question in cellulose chemistry. Since the discovery of the crystallographic differences of native cellulose I and mercerized cellulose II, which would also be the crystal structure present in regenerated cellulose, it remained disputed how the transition from nanocrystals with parallel to those with antiparallel cellulose molecules would occur. A certain mobility of the long chain molecules is required and, in the case of mercerization, enabled by swelling with sodium hydroxide. Two options have been proposed for the rearrangement of the cellulose molecules. Chanzy and Roche (CR) suggest a folding mechanism of the chains within each crystal, whereas Kolpak and Blackwell (KP) assume migration of molecules from one crystal to another.

The challenge to decide whether the CR or the KB model is correct involves "marking" the molecules of one nanocrystal with respect to another one. The marker used in this work is an exchange of hydrogen by deuterium within the entire nanocrystal. The technique that deuterium exchange marking works for is neutron (fiber) diffraction. Deuteration can be achieved for bacterial cellulose: the bacterium *Glucanacetobacter xylinus* manages to live and grow in a fully deuterated environment. The authors further have developed a technology to prepare well-aligned oriented samples based on nanocrystals so that the fiber symmetry of the samples is ensured. The crucial sample is the one where a 50:50 mix of hydrogenated and deuterated nanocrystals is present; fully protonated and fully deuterated samples are measured for comparison.

For mercerization, a deuterated solution of NaOH (thus, NaOD + D₂O) was used. The authors carefully checked that the level of OH/OD exchange is similar in all samples; I think that could be stressed even more as the contrast technique is very sensitive to that.

RESPONSE: We agree with the reviewer. We have included further details (highlighted in yellow and with revision tracking) about the preparation and characterization of the samples, stressing the importance of achieving similar levels of OH/OD exchanges. We have also added a methods section, within which sample preparation is outlined in more detail.

For the logic of the work presented here, Fig. 1 is central. It is used in two ways. First, considering only the figure caption, the color scheme is more or less redundant as blue and red stand for two opposite orientations of the (chiral) cellulose molecules, depicted also by arrows.

RESPONSE: We agree with the reviewer. We have removed the color scheme from this figure.

Second, in the main text, it is stated that blue and red may also signify protonated and deuterated chains in nanocrystals. Here, Fig. 1 does not show the full picture though. The starting point of the mixed sample before mercerization would be characterized by equal amounts of protonated and deuterated (thus, blue and red) nanocrystals with arrows up and down. If, in a simplified approach, the combination of chains of opposite polarity was only happening between neighboring nanocrystals of opposite polarity four different combinations of the blue and red arrows would be possible. Please refer to my attached sketch of the situation. Eventually, only 50 % of the cellulose II nanocrystals would still be

mixed red and blue then, 25 % each would be just red or blue, i. e., fully protonated or fully deuterated, respectively. [Only of the chains were mobile beyond nearest neighbors, more complex combinations would also occur; see the rightmost figure in my sketch.] My feeling would be that the difference in Bragg reflection intensity ratios would be smaller than in Table 1. I cannot fully exclude that a decision between the CR and the KB model may no longer be possible on the basis of the measured data. I suggest that the authors make the according calculations to decide.

RESPONSE: We have considered the more complete permutations, including two bounding case which we think represent the extremes of possibility with the KB model. The difference in Bragg intensities is still is significant enough to differentiate between CR and KB. We have revised the manuscript to include the calculations suggested by the reviewer. We have also included more details on how the calculations were done (highlighted in blue) in the Results and Methods section, including three additional reference (15,16,17) and a Table 2. Table 1 has been updated. We have also added a Figure (3) that is based on the reviewer's sketch. We have renamed the original Figure 3 as Figure 4..

It is still valid though that the CR model would lead to the CDH pattern being an average of CH and CD. Still, I am not sure whether the corrected KB pattern would still be as distinctively different as Table 1 suggests. The nice additional argument via the pronounced small-angle scattering of the mixed sample is also valid, however, the data are not sufficiently quantitative here.

RESPONSE: We agree with the reviewer that the small-angle data are not sufficiently quantitative here, and we make that point in the text.

A last remark concerns the figures. It might be easier for the reader if, instead of ABCD, either the abbreviations of the model (CR and KB for Fig. 1) were used, in Fig. 2 X-ray, CH, CD, CDH may guide the reader, or in Fig. 3 CD+CH and CDH, respectively.

RESPONSE: We agree with the reviewer and we have replaced ABCD in the figures with the abbreviations for the models.

In conclusion, I hope that the innovative approach presented here in order to reveal the molecular origin of mercerization still works after the - in my opinion - necessary recalculation for the KB model.

RESPONSE: We thank the reviewer for the insightful recommendations.

Reviewer #2 (Remarks to the Author):

The manuscript describes a new approach to resolve a long-standing issue of the physical process behind mercerization of cellulose I to cellulose II. The authors were able to generate (sufficient) batches of hydrogenated and deuterated cellulose for a neutron diffraction experiment. By combining H-containing cellulose fibrils with D-containing cellulose fibrils, the authors perform a neutron diffraction experiment to discern between two models (CR vs KB) that describe the transition between the crystallographic state I and II. The approach is novel and the topic is relevant to bio-polymer physics in general, with cellulose as the most bio-abundant polymer.

The methodology is novel and clear but in interpreting the data, the authors can provide the reader with some assistance in reading the figures. The Q values of the reflections are assumed common knowledge as are the inter-domain and intra-domain ordering of the polymers.

RESPONSE: We have including the Q (as distances in real space and reciprocal space) for the two Bragg reflections. We have included a brief description of the unit cell and ordering of polymers within it, in the Results and Methods sections (highlighted in grey). These Bragg reflections do not reflect inter-domain ordering (except the very low angle feature, which we have removed from the discussion).

The authors point out:

267: A detail of the lower angle region shows an equatorial intensity feature (indicated by arrow D) for CDH-cellulose II that does not appear for CH-cellulose II or CD-cellulose II.

The authors relate this 'strong feature' to small-angle scattering on nanoscopic crystalline domains: the required contrast (H-domain vs D-domain) would then also appear in their cellulose I samples, prior to mercerization, but such (supporting) data is not shown. Indeed, the observation itself calls for a SANS experiment.

RESPONSE: As discussed in response to the other reviewer, we agree that the small-angle data are not sufficiently quantitative here, and we make that point in the text, and de-emphasize the significance of this feature. However, this is a great question. When we prepared our samples it took a tremendous amount of time and effort (several months). We first of all prepared hundreds of thin strips of aligned cellulose I nanocrystals, each a few millimeters in width. We checked each of those strips with X-rays. Then we mercerized them to obtain thin strips of cellulose II. We checked these strips again with X-rays and FTIR to ensure complete conversion to cellulose II and fibrinogen removal. We then assembled them into a parallel bundle which contained sufficient volume for neutron diffraction studies (neutron beams are relatively weak). We did this three times for CH cellulose II, CD cellulose II and CDH cellulose II samples.

However, we didn't have enough material or time to prepare bundles of strips for neutron studies of CH-cellulose I, CD-cellulose I and CDH cellulose I, although I agree it would have been really interesting to do.

Does the Q-value of the reflection corresponding to the domain size in an integer fashion? Figure 7c in [13] shows a similar darkening close to the beam stop, suggesting a similar ordering in 100%-D cellulose II, which would be in contradiction with the explanation given in the current manuscript. In the (current) manuscript, figure 2C shows a very faint darkening at the same Q value. Although one could discard such darkening as too faint, the variations in table 1 between integrated intensities are of the order of their inaccuracies (assumed to be stated in parentheses behind the intensity value).

RESPONSE: As discussed in response to the other reviewer, we agree that the small-angle data are not sufficiently quantitative here, and we make that point in the text, and de-emphasize the significance of this feature. However, again this is a great question. The sample in Figure 7c in [13] is a sample of hydrogenous cellulose that has been soaked in D₂O, so that labile and accessible H has been replaced by D. However, H within the cellulose fibrils is mostly not replaced by D. The deuteration therefore

mostly occurs in the regions between cellulose fibrils and introduces contrast between fibrils and the less ordered matrix in that way.

some minor remarks/observations:

line 85: mixture of hydrogenated and deuteration nanocrystals,  deuterated?

line 243: pattern

RESPONSE: We have made these corrections.

The main conclusion of the manuscript is that experiment supports the C&R model rather than the K&B model, so polymer folds onto itself rather than intermixing with other molecules through diffusion. Table 1 shows that CDH-II is indistinguishable from the addition of CH-II and CD-II. The K&B process would result in a doubling of the cell parameter for the CHD-II system. Does the calculated intensity of table I correspond to the new cell reflection? Summarizing, the manuscript provides new insights in an important bio-chemical process, but can easily be improved in discussing the interpretation in a more quantitative manner.

RESPONSE: We agree with the reviewer that a more quantitative discussion will improve the manuscript and we have tried to do so. We have included more discussion (highlighted in grey). The calculated intensity in table I corresponds to the same reflection from the same size and type of unit cell, but with different amounts proportions of H and D polymers. The CR and KB processes would not result in changes of unit cell parameters for the CHD-II system, they could result in increases in the size of the crystalline domains (which would contain more unit cells).

Reviewers' Comments:

Reviewer #1:

Remarks to the Author:

I read the revised manuscript with pleasure. The authors have addressed all issues raised by the two reviewers. I am particularly impressed by the full set of calculations added in the analysis of the KB model for mercerization. It is a very good idea to define the two limiting scenarios for chain exchange between cellulose nanocrystals and to quantify them. I am very happy to see that the main conclusion remains valid. In my opinion, the paper will be considered as a milestone for the full understanding of the cellulose I to II conversion.

It was absolutely necessary to increase the length of the manuscript to accommodate the additional explanations and discussions. I explicitly do not recommend any shortening here.

Reviewer #2:

Remarks to the Author:

The authors have properly responded to each of the points I raised in the rebuttal and made the relevant changes in the manuscript text. The main argumentation that the experimental data supports the C&R model stands.

Following the remarks of the other reviewer, the authors have modified figure 1 and added figure 3, for clarification of the different 'mixing schemes' in mercerization. The K&B model of converting from cellulose I to II is depicted as a doubling of the cell parameter in the "50% CDH Cellulose II" sub-figure, while the calculation only takes the increase in intensity into account. It would clarify the argumentation if the authors would elaborate on the fact that the "50% CDH part" of cellulose 2 does not appear as a second/separate crystal structure and therefore at $Q/2$ for the 1-10 plane in the fiber diffraction. Maybe a mention of the aspect ratio of the nano-crystals or depicted domains would help to clarify.

Final Point-by-point response

Reviewer #1 (Remarks to the Author):

I read the revised manuscript with pleasure. The authors have addressed all issues raised by the two reviewers. I am particularly impressed by the full set of calculations added in the analysis of the KB model for mercerization. It is a very good idea to define the two limiting scenarios for chain exchange between cellulose nanocrystals and to quantify them. I am very happy to see that the main conclusion remains valid. In my opinion, the paper will be considered as a milestone for the full understanding of the cellulose I to II conversion.

It was absolutely necessary to increase the length of the manuscript to accommodate the additional explanations and discussions. I explicitly do not recommend any shortening here.

Response: We thank the reviewer for helping us to make this a stronger manuscript.

Reviewer #2 (Remarks to the Author):

The authors have properly responded to each of the points I raised in the rebuttal and made the relevant changes in the manuscript text. The main argumentation that the experimental data supports the C&R model stands. Following the remarks of the other reviewer, the authors have modified figure 1 and added figure 3, for clarification of the different 'mixing schemes' in mercerization.

The K&B model of converting from cellulose I to II is depicted as a doubling of the cell parameter in the "50% CDH Cellulose II" sub-figure, while the calculation only takes the increase in intensity into account. It would clarify the argumentation if the authors would elaborate on the fact that the "50% CDH part" of cellulose 2 does not appear as a second/separate crystal structure and therefore at $Q/2$ for the 1-10 plane in the fiber diffraction.

Response: We agree. We have added the following sentence in the sub-figure caption (Figure 3) "The size of the unit cells for 25% CH and 25% CD and the averaged cell for 50% CDH are the same. Therefore, the position of intensities in the diffraction pattern from a sample containing mixtures of these different unit cells would coincide. We do not expect additional reflections."

Maybe a mention of the aspect ratio of the nano-crystals or depicted domains would help to clarify.

Response: We agree. We have added the following sentence in the discussion section when referring to the nano-domains "We note that their aspect ratio will be different from the crystal domains in cellulose I, having decreased along the length of the cellulose chain direction and increased in the plane normal to this direction"